# Re-Analysis and Additional Information Needed to Inform Conclusions. Comment on Halenova et al. Deuterium-Depleted Water as Adjuvant Therapeutic Agent for Treatment of Diet-Induced Obesity in Rats. *Molecules* 2020, *25*, 23

**DOI:** 10.3390/molecules27165186

**Published:** 2022-08-15

**Authors:** Colby J. Vorland, Xiwei Chen, Daniella E. Chusyd, Luis M. Mestre, Stephanie L. Dickinson, David B. Allison, Andrew W. Brown

**Affiliations:** 1Department of Applied Health Science, Indiana University School of Public Health-Bloomington, Bloomington, IN 47405, USA; 2Department of Epidemiology and Biostatistics, Indiana University School of Public Health-Bloomington, Bloomington, IN 47405, USA

We were interested to read the report by Halenova et al. [1], who tested the effect of deuterium-depleted water (DDW) on obesity-related outcomes using a rat model of diet-induced obesity (DIO) vs. control rats. Within each of these groups, the authors report randomizing half to DDW and the other half to MilliQ-filtered water for three weeks. The authors concluded that DDW mitigates DIO. We have concerns about how the data were analyzed based on treatment assignments, and question some of the reported results.

Regarding the statistical analyses, the authors report using Student’s *t*-test for their statistical comparisons, which has several potential concerns. First, whether animals were individual- or group-caged is not clear. The authors reference an unrelated prior study [2] for details of animals and housing in which they state that animals were housed in groups of five. The treatment in this study ([1]) was provided in water, which is reasonable to assume was provided to the *cage*, not each *individual rat*. If the same protocol was followed, then each condition (n = 10) was housed in two cages (n = 5 in each cage). The units of analysis are then not independent observations as assumed in a Student’s t-test. The correlation of animals within cage should be taken into consideration in the analysis, or the assessment of treatment effects may result in an inflated type I error rate [3,4,5,6]. If the animals were randomized by cage, this adds additional considerations as inferences are based on two units of assignment per group, not ten [7,8]. In addition to group-housing altering standard errors and degrees of freedom for any between-group tests, the cage-level treatment makes any food or water data represent averages across rats resulting from two independent measurements per treatment, not ten. Both scenarios (randomizing by cage and treating animals at the cage-level) need to be accounted for in power calculations to estimate sample size [6,7,9]; a sample size calculation was not reported by the authors. As in any power calculation, power is dependent on variability in the number of independent observations. By only having two independent observations for each condition (i.e., two cages for DDW and two for MilliQ-filtered water), this limits the degrees of freedom; there are only two independent groups per treatment. When considering group-housed animals, the estimate of the variability is dependent on the “design effect” (a function of the average number of rats per cage and intraclass correlation coefficient), details of which have been well documented elsewhere [4,7].

Second, additional information is needed to inform the conclusion about whether DDW has an anti-obesity effect. Per the authors’ Figure 1A, the percent weight gain relative to baseline weight at week 0 appears to be greater in the control (standard chow) + DDW group as compared to the control + MilliQ at week 8. In Figure 1B, however, body weight index (BWI; calculated as grams/cm^2^) in the control + DDW group is on average lower than both DIO groups. Plausible explanations for this include: (1) there were group imbalances at baseline so that the control rats had more growth potential, or (2) the animals had marked linear growth after the introduction of DDW so that BWI was lower than the other groups, despite the relatively higher change in weight. A more mundane explanation is that the treatments of animals or figures were misidentified. Because weight in neither grams nor lengths is reported, the reader is unable to determine which is more likely. Further, the authors report that the control + DDW group increased water intake compared to the control + MilliQ group, meaning that changes in weight are confounded by these differences.

Per their paper, the authors note “The experimental data used to support the findings of this study are available from the corresponding author upon request”. Unfortunately, we attempted to resolve our questions by contacting the authors for their data, but attempts by us and the journal to reach the authors were unsuccessful. We, therefore, request that the authors provide clarification to our inquiry, including clarifying whether rodents were group-treated and, if so, re-analyzing their results taking into consideration the hierarchical structure of the data, as well as reporting baseline body weights for each group. Per the data availability statement, we also request that raw data are shared so we and others may perform re-analyses. We furthermore request that the authors report p-values to exact values per reporting guidelines, instead of, e.g., “*p* < 0.05”, to aid interpretation [10].

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
