# Peer review of "Re-Analysis and Additional Information Needed to Inform Conclusions. Comment on Halenova et al. Deuterium-Depleted Water as Adjuvant Therapeutic Agent for Treatment of Diet-Induced Obesity in Rats. Molecules 2020, 25, 23"

_molecules, 2022, doi:10.3390/molecules27165186_

Round 1
Reviewer 1 Report
The issues raised by the authors appear valid, and it seems appropriate that the comment be published without editing or revising. The authors of the original paper should be given an opportunity to respond, and either their response or a note of their decline to respond should accompany the comment. Many of the questions raised in the comment are straightforward and could be addressed in a response.
Author Response
We thank the reviewer for their time and expertise.
Reviewer 2 Report
I read with interest both the paper by Halenova et al. T.; Zlatskiy, I.; Syroeshkin, A.; Maximova, T.; Pleteneva, T. Deuterium-depleted water as adjuvant therapeutic agent for treatment of diet-induced obesity in rats, and the comment by Vorland et al.
I have also read the responses of the authors of the original article. to the "Comment On: Deuterium-Depleted Water as Adjuvant Therapeutic Agent for Treatment of Diet-Induced Obesity in Rats"
Vorland et al. highlight two issues requiring more detailed information.
- First, the authors of the comment have reservations about the statistical analysis. In fact, it is described quite briefly, which, however, was probably satisfactory for the reviewers. Without deciding whether the statistical analysis was correct, it should be noted that the arguments of Vorland et al. are not convincing in this case. The articles they cite refer to completely different experimental models and do not seem to apply here.
- Sainani [1] states quite well what is meant by the term "correlated observations": "Correlated data arise when pairs or clusters of observations are related and thus are more similar to each other than to other observations in the dataset." And next "Observations may be related because they come from the same subject—for example, when subjects are measured at multiple time points (repeated measures) or when subjects contribute data on multiple body parts, such as both eyes, hands, arms, legs, or sides of the face. Observations from different subjects also may be related—for example, if the dataset contains siblings, twin pairs, husband-wife pairs, control subjects who have been matched to individual cases, or patients from the same physician practice, clinic, or hospital. Cluster randomized trials, which are performed to assign interventions to groups of people rather than to individual subjects (for example, schools, classrooms, cities, clinics, or communities), also are a source of correlated data because subjects within a cluster will likely have more similar outcomes than subjects in other clusters." Clearly, such a situation does not occur in the described experimental model.
- The reference to the article by Murray et al.[2] - concerns the so-called Group-randomized trials (GRTs), which are comparative studies as opposed to the experimental study in question. Muray et al.[2] wrote: "Group-randomized trials (GRTs) are comparative studies designed to evaluate interventions that operate at a group level, manipulate the physical or social environment, or cannot be delivered to individuals.1 Examples include school-, worksite-, and community-based studies designed to improve the health of students, employees, and residents, respectively."
- The second remark concerns a minor but significant problem, the inverse trend of weight and BWI changes. In this case, the comment by Vorland et al. seems legitimate and worth answering.
However, the mentioned response to the "Comment On: Deuterium-Depleted Water as Adjuvant Therapeutic Agent for Treatment of Diet-Induced Obesity in Rats" is quite superficial and do not answer the authors' concerns.
- Sainani, K. The importance of accounting for correlated observations. PM R 2010, 2, 858-861, doi:10.1016/j.pmrj.2010.07.482.
- Murray, D.M.; Varnell, S.P.; Blitstein, J.L. Design and analysis of group-randomized trials: a review of recent methodological developments. Am J Public Health 2004, 94, 423-432, doi:10.2105/ajph.94.3.423.
Author Response
We thank the reviewer for their thoughtful comments, and for their acknowledgement that our second concern deserves a response from the authors. We have expanded our discussion below and added references to our commentary to address the reviewer’s note that our first concern was “not convincing in this case.” The statistical and design issues we address are similar whether the study involves humans in groups (e.g., classrooms), pigs in pens, or rats in cages. However, our initial choice of references emphasized humans. We have therefore added references that make the issues in animal research more clear.
1) Importance of accounting for correlations in statistical analysis:
Subjects (humans or rats) that are related in that they interact within intact social groups have observations that may be correlated. The animals interact, eat, and drink from the same sources, and influence one another’s behavior in other ways. Their observations thus cannot be assumed to be independent. Any change within a cage may affect all animals in the cage, and thus may affect all observations. As an example (see ref #1 below):
“… if we accidentally give twice the concentration of a drug in the drinking water, then all animals in a cage that share a water bottle will be affected.”
Note that the treatment in Halenova et al. was given in water.
As another discussion of clustering/correlated observations in animal research (see ref #2 below):
“Depending on the experiment, we recognize that it is not always possible to single-house mice. Our review showed that scientists often analyze clustered observations using methods that mathematically function under the assumption of data independence (student T-, Mann-Whitney, One-/Two-way ANOVAs), without implementing statistics for intra-class (‘intra-cage’) correlated (ICC) cage-clustered data (Multivariable linear/logistic, Marginal, Generalized Estimating Equations, or Mixed Random/Fixed Regressions). The ICC describes how units in a cluster resemble one another, and can be interpreted as the fraction of the total variance due to variation between clusters.”
This is why it is critically important to know how many animals were housed within a cage. Halenova et al. referenced a previous study in their methods for details on the animals and how they were housed, in which they write that animals were housed five to a cage. Thus, we requested confirmation whether this occurred for the animals used in this paper, and if so, that they account for this in their analysis. They report using Student’s t-tests in their analysis, which does not account for the nonindependence of observations.
2) Why specialized statistical approaches are needed for group randomized trials:
The scenario above assumes that animals were allocated individually and then treated together as a group within cages. However, a more severe error occurs if the animals were allocated to treatments as a group and treated together as a group within their cages. For example, before the start of the experiment, if animals were initially housed together in their four cages with n=5 each (as suggested by the reference that Halenova et al. provide) and those cages were then assigned to either the treatment or the control, this is a group randomized trial. The principles of group randomized trials applies to animals (ref #2 below) as well as humans as discussed in the Murray et al. reference. In this scenario, the cage is the unit of assignment, which results in n=2 cages for treatment and control. In other words, there are only 2 independent units of assignment per treatment instead of 10, and the Student’s t-test is not an appropriate method to analyze the results as the authors have. We have written about the invalidating consequences of misanalysing group randomized studies in this context in animal research (See Error #5 in [3] below), and more generally [4-6 below]. If animals were allocated in this manner, failing to account for this increases the probability of a false positive finding, as we note in our letter.
We have added an additional reference (#3 below) to our manuscript for additional background for readers that includes a discussion of this error in context of group randomization of animals.
- Lazic, S. E., Clarke-Williams, C. J., & Munafò, M. R. (2018). What exactly is ‘N’in cell culture and animal experiments?. PLoS biology, 16(4), e2005282.
- Basson, A. R., LaSalla, A., Lam, G., Kulpins, D., Moen, E. L., Sundrud, M. S., Miyoshi, J., Ilic, S., Theriault, B. R., Cominelli, F., & Rodriguez-Palacios, A. (2020). Artificial microbiome heterogeneity spurs six practical action themes and examples to increase study power-driven reproducibility. Scientific Reports, 10(1). https://doi.org/10.1038/s41598-020-60900-y
- Vorland, C. J., Brown, A. W., Dawson, J. A., Dickinson, S. L., Golzarri-Arroyo, L., Hannon, B. A., ... & Allison, D. B. (2021). Errors in the implementation, analysis, and reporting of randomization within obesity and nutrition research: a guide to their avoidance. International Journal of Obesity, 45(11), 2335-2346.
- Brown, A. W., Li, P., Bohan Brown, M. M., Kaiser, K. A., Keith, S. W., Oakes, J. M., & Allison, D. B. (2015). Best (but oft-forgotten) practices: designing, analyzing, and reporting cluster randomized controlled trials. The American journal of clinical nutrition, 102(2), 241-248.
- George, B. J., Beasley, T. M., Brown, A. W., Dawson, J., Dimova, R., Divers, J., Goldsby, T. U., Heo, M., Kaiser, K. A., Keith, S. W., Kim, M. Y., Li, P., Mehta, T., Oakes, J. M., Skinner, A., Stuart, E., & Allison, D. B. (2016). Common scientific and statistical errors in obesity research. Obesity, 24(4), 781–790. Portico. https://doi.org/10.1002/oby.21449
- Brown, A. W., Altman, D. G., Baranowski, T., Bland, J. M., Dawson, J. A., Dhurandhar, N. V., Dowla, S., Fontaine, K. R., Gelman, A., Heymsfield, S. B., Jayawardene, W., Keith, S. W., Kyle, T. K., Loken, E., Oakes, J. M., Stevens, J., Thomas, D. M., & Allison, D. B. (2019). Childhood obesity intervention studies: A narrative review and guide for investigators, authors, editors, reviewers, journalists, and readers to guard against exaggerated effectiveness claims. Obesity Reviews, 20(11), 1523–1541. Portico. https://doi.org/10.1111/obr.12923
Reviewer 3 Report
This comment on "Deuterium-3 Depleted Water as Adjuvant Therapeutic Agent for 4 Treatment of Diet-Induced Obesity in Rats. " put forward two concerns. First, as the authors of the article mentioned above did not clear the protocols, it is hard to figure out whether the statistical analysis method used in the original article "Student’s t-test" is proper. Second, due to the lack of some key data such as "baseline body weights", some of the results are difficult to explain.
I am very much enjoyed reading this comment, for its well edited, clean and pristine. Sorry for no significant comments. I believe the first concern towards the statistical analysis method is vital to identify whether the result is proper and the second concern may be helpful for other readers if solved.
Author Response

(The authors gave the same response as above.)

Reviewer 4 Report
In the present comment, Colby J. Vorland and colleagues discussed the report by Halenova et al. (Halenova, T.; Zlatskiy, I.; Syroeshkin, A.; Maximova, T.; Pleteneva, T. Deuterium-depleted water as adjuvant therapeutic agent for treatment of diet-induced obesity in rats. Molecules 2020, 25, 23) who tested the effect of deuterium depleted water (DDW) on obesity-related outcomes using a rat model of diet-induced obesity (DIO) vs. control rats; in this paper the authors concluded that DDW mitigates DIO. Vorland and colleagues have concerns about how the data were analysed based on treatment assignments, and question some of the reported results. Overall, I think that the from a scientific point of view the observations are timely and they could also be sharable. Indeed, in the fascinating field of research, I feel that any attempt to improve research represents a stimulus for science community. However, I believe that some of the questions posed by the authors of the letter were deeply discussed with the reviewers during the manuscript submission process and the last request in the comment “We therefore request that the authors provide clarification to…” merit, of course, an adequate right of reply by authors of original manuscript, even if the eventual reanalysis is an exclusive prerogative of the same authors.
As a matter of fact, the authors of the "Comments" could reproduce their research on the topic of manuscript, taking into account all interesting comments and suggestions and, only then, I feel that the scientific debate it would be more appropriate and definitely constructive.
Author Response
We thank the reviewer for their comments, and agree that a response to our questions from the authors is warranted. While performing a separate study in the future is recommended to establish the reproducibility of any finding, that does not change the importance of ensuring the rigor and validity of the present study.
Reviewer 5 Report
This comment discusses about the paper published by Halenova et al. (10.3390/molecules25010023).
In my opinion, the authors raised several important concerns about the paper of Halenova et al.
I have only a few suggestions:
- First of all, the authors should justify the reason for the delay of their comment, considering that the article has been published more than 2 years ago;
- Moreover, considering that it is a comment, I suggest removing the funding information. It is not clear how a grant may support a comment.
Author Response
To the first point, we attempted to quickly resolve our questions immediately after it was published. The paper was published in December of 2019, and we noticed it and began discussing it within our group that month. Over the course of the next nearly six months, we contacted each of the authors on the paper several times to request raw data to which they did not reply. The journal also contacted them and was unable to get a response. Given the gravity of the potential errors, we submitted a letter in early 2021. It took approximately one year for the journal to receive a reply from the authors and send our letter out for review. Although these details may be interesting for the sake of discussion, we do not believe they add to the discourse about the rigor and validity of the present study, so we prefer to forego describing them in our correspondence.
To the second point, our funding is relevant because the grants include support for efforts to enhance the rigor, reproducibility, and transparency of obesity and related sciences.
Round 2
Reviewer 2 Report
I am still not convinced by the author's arguments, but I understand their reasoning. I think it would be best if the original article's authors responded to these comments, especially as they are the only ones who have access to the complete experimental data.
Author Response

(The authors gave the same response as above.)
